# Acute and Reproductive Toxicity Evaluation of Ormona^®^ SI and Ormona^®^ RC—Two New Nutraceuticals with Geranylgeraniol, Tocotrienols, Anthocyanins, and Isoflavones—In Adult Zebrafish

**DOI:** 10.3390/ph15111434

**Published:** 2022-11-19

**Authors:** Clarice Flexa da Rocha, Camila de Nazaré Nunes Flexa, Gisele Custodio de Souza, Arlindo César Matias Pereira, Helison de Oliveira Carvalho, Aline Lopes do Nascimento, Natasha Juliana Perdigão de Jesus Vasconcelos, Heitor Ribeiro da Silva, José Carlos Tavares Carvalho

**Affiliations:** 1Laboratory of Drugs Research, Biology and Healthy Sciences Department, Pharmacy Faculty, Federal University of Amapá, Macapá 68902-280, Brazil; 2Pos-Graduation Program in Pharmaceutical Innovation, Biology and Health Science Departament, Pharmacy Course, Federal University of Amapá, Macapá 68902-280, Brazil; 3School of Pharmaceutical Sciences of Ribeirão Preto, University of São Paulo (FCFRP-14USP), São Paulo 14040-903, Brazil; 4University Hospital of the Federal University of Amapá, Macapá 68902-336, Brazil

**Keywords:** isoflavones, tocotrienol, geranylgeraniol, anthocyanins, phytoestrogens

## Abstract

The zebrafish is a popular organism to test the toxicity of compounds. Here, we evaluate the acute and reproductive toxicity of Ormona SI^®^ (OSI) and RC^®^ (ORC), two herbal products developed for menopausal women with tocotrienols, geranylgeraniol, isoflavones, and anthocyanins. The acute toxicity was evaluated by behavioral alterations, lethality, and tissue changes (intestine, liver, kidney) after oral treatment with high product doses (500, 1000, and 2000 mg/kg). The reproductive toxicity was evaluated after 21 days of oral treatment with OSI and ORC at 200 mg/kg. Our results show that the LD_50_ could not be assessed due to the low mortality rate even with the highest dose; the behavioral alterations were not different from those of the group treated only with the vehicle (2% DMSO). The tissue changes were minor in OSI and more severe in ORC at the highest (2000 mg/kg) dose, while no tissue abnormality was observed at 500 mg/kg. In the reproductive assessment, continuous treatment could decrease the maturation of the reproductive cells, which also significantly decreases the egg spawning. This effect was attributed to the estrogenic activity of the isoflavones. In conclusion, the acute toxicity analysis shows that the products did not elicit lethal or sublethal effects observed in the model when used up to 500 mg/kg. Regarding the reproductive toxicity, decreased fertility was observed, which was expected due to the presence of isoflavones (phytoestrogens). Considering that the product is intended for menopausal and postmenopausal women, the presence of isoflavones is beneficial. Further studies should be performed to corroborate these results in mammals.

## 1. Introduction

Ormona^®^ is a herbal product for menopausal women whose bioactive compounds are derived from annatto oil granules (*Bixa orellana*, 42.5%), isoflavone-rich fractions (20.8%) of soybean germ (*Glycine max*) in Ormona^®^ SI (OSI), or red clover (*Trifolium pratense*) in Ormona^®^ RC (ORC), and a standardized extract of açaí (*Euterpe oleracea*, 10%). The annatto oil has 10% tocotrienols (mainly the δ isomer) and 28% geranylgeraniol. In the isoflavone-rich fraction of OSI, there is genistein (1.08%), genistin (0.16%), daidzein (3.84%), glycitein (3%), among other isoflavones; in the isoflavone-rich fractions of ORC there is daidzein (5.4%), glycitein (2.3%), and genistein (3.6%). Finally, the standardized extract of açaí is composed mainly of anthocyanins (60%), with a predominance of cyanidin 3-O-glucoside [1].

Isoflavones are compounds with structural similarity to 17β-estradiol commonly found in soy and red clover. Due to this similarity, some isoflavones have estrogenic activity and are reported to decrease some menopausal symptoms without any safety issues at the tested doses [2,3,4,5,6]. The annatto granules in Ormona^®^ SI are the same as in Chronic^®^, which has been shown to improve the blood-lipid profile and bone calcium content in vivo [7], in accordance with studies reporting an improvement of blood-lipid profile in humans with tocotrienol treatment [8,9]. Annatto tocotrienols are safe for postmenopausal women [10] and significantly improve osteopenia in treated menopausal patients compared to a placebo group [11]. Meta-analyses of clinical trials have also reported that anthocyanins improve the blood-lipid profile and systemic inflammation, indicating a decreased risk for cardiovascular diseases [12,13].

Toxicity studies are essential in developing new products to assess if they are safe at the consumed quantity. There are several approaches to assessing the potential toxicity of such compounds; a popular organism model in toxicology studies is zebrafish [14,15,16]. Hence, this study aimed to evaluate the acute and reproductive toxicity of Ormona^®^ SI and Ormona^®^ RC using zebrafish.

## 2. Results

### 2.1. Acute Toxicity

Treating adult zebrafish with high OSI and ORC doses caused behavioral changes, such as increased swimming activity, tail tremors, and bottom immobility. Of the five fish treated in each group, two died in the groups treated with the highest doses of OSI and ORC (1000 and 2000 mg/kg). No death occurred in the group treated with OSI or ORC at 500 mg/kg, and only minor behavioral alterations were observed. However, the same behavioral alterations occurred in the group treated only with the vehicle, meaning that they were not necessarily caused by the OSI or ORC (Table 1). Due to the low mortality observed, even with high doses, it was impossible to calculate the LD_50_.

Besides behavioral alterations and deaths, we sought to observe possible sublethal organ toxicities. Hence, histopathological analysis was performed on the animals’ kidneys, livers, and intestines (Figure 1). Histopathology can show organ-specific toxicities that cannot be observed externally caused by natural compounds.

Despite being different from the DMSO group, no severe tissue change was observed in animals treated with OSI (Figure 2) since the HCI was always inferior to 10. ORC had more severe, albeit reversible, tissue changes caused by the highest dose of 2000 mg/kg (Figure 2). Notably, neither OSI nor ORC induced significant toxicity at 500 mg/kg. Figure 1 shows representative tissue changes counted for quantification.

The kidney is used to excrete metabolites through the urine and regulates the osmotic and acid–base balance of animals [17]. According to [18], this organ is susceptible to toxic compounds. The most frequent tissue change observed was soft hyaline degeneration (Figure 1E,F). As observed in the other organs, the treatment induced more significant tissue changes only in the highest dose of ORC, while no toxicity was observed with the lowest dose (Figure 2).

### 2.2. Reproductive Toxicity

During the pretreatment, the spawning occurred in regular intervals, with unaltered fertility. After treatment, it was observed that the untreated group had a fertility rate similar to that of the control group treated only with the vehicle of 2% DMSO. The spawning in animals treated with OSI at 200 mg/kg decreased drastically compared to the control groups (Figure 3). ORC at 200 mg/kg could also decrease the spawning rate, although the difference was not statistically significant compared to the control (Figure 3).

None of the eggs from treated parents (OSI and ORC) survived (Figure 3), so it was impossible to assess malformations in the embryos. No deaths or malformations were observed in the control groups—such as cardiac edema, tail malformation, scoliosis, yolk edema, and growth retardation. A possible way parental treatment could induce toxicity in the embryos would be by changing the quality of the reproductive cells [19]. Hence, we sought to assess any tissue change in the gonads of parent fishes (Figure 4).

It was observed that the treatment with OSI and ORC increased the percentage of early-stage reproductive cells whilst decreasing the percentage of mature reproductive cells in female and male animals. In females, the percentage of cortical alveoli oocytes was statistically higher in OSI-treated animals, while the percentage of early vitellogenic oocytes and late mature oocytes was lower in female animals. For ORC-treated animals, only the frequency of late mature oocytes was affected (Figure 5).

In males, OSI-treated animals had an increased percentage of spermatogonia and decreased percentage of spermatozoa. Again, ORC did not affect the frequency of early-stage reproductive cells, but the frequency of spermatozoa decreased substantially (Figure 6).

## 3. Discussion

Ormona^®^ is a herbal product for menopausal women. Considering that dyslipidemia is a common complication of aging, we previously evaluated the effect of this product on serum biochemistry and atherogenesis in dyslipidemic rats. We showed that Ormona^®^ SI (OSI) and Ormona^®^ RC (ORC) significantly prevented the increase in total cholesterol, LDL, triglycerides, and glucose while preventing HDL decrease. The treatment also prevented atherogenesis in the aorta and decreased the levels of AST, ALT, total bilirubin, urea, and creatinine in the serum [1].

The product is composed of annatto oil granules (42%, of which 28% is geranylgeraniol and 10% tocotrienols, analyzed through GC-MS), 10% standardized açaí extract rich in anthocyanins, mainly cyanidin 3-O-glucoside (60% of the extract), and, finally, an isoflavone-rich fraction (20.8%). From the isoflavone-rich fraction, we could identify genistein (1.08% in OSI, 3.6% in ORC; *w/w* dry weight), daidzein (3.8% in OSI, 5.4% in ORC), and glycitein (3% in OSI, 2.3% in ORC). Other peaks could not be identified in the isoflavone fraction but presumably could be formononetin and biochanin A (Figure 7). Considering the presence of isoflavones, we sought to evaluate OSI and ORC reproductive toxicity in zebrafish.

According to [18], it is common to observe histopathological changes in the intestine of zebrafish after oral treatment since this is the first organ of contact. The intestine of zebrafish is relatively simple, formed by enterocytes, goblet cells, and some inflammatory cells [21], and it is the organ where the macronutrients are absorbed [22]. Here, the most frequent tissue change observed in this organ was hypertrophy of the epithelial cells and dilation of vessels (Figure 1A,B). In the highest doses of ORC, detachment of the lamina propria was observed as well.

The liver of zebrafish is accountable for metabolizing drugs and other xenobiotics via the action of cytochrome P450. In this aspect and its structure, it is very similar to that of humans [23,24]. The most frequent tissue change observed in the animals’ liver was cytoplasm vacuolization (Figure 1C,D), frequently reported in the literature [17,18,22,23]; this feature could indicate a reduction of glycogen reserves or lipid accumulation, which in excess could impair organ function. Our results show that OSI caused only soft tissue changes, while ORC’s highest dose had a more severe effect at the highest dose, with the occurrence of hepatocyte degeneration and hyperemia. Again, both formulations were innocuous at 500 mg/kg (Figure 2).

The earlier reported decreased rate of spawn and survival of the embryos is explained by the decreased degrees of maturation in reproductive cells. The effect of Ormona^®^ on the maturation of reproductive cells confirms the estrogenic activity from the isoflavones in its composition (20%). The differences between Ormona SI^®^ and Ormona RC^®^ can be attributed to the different profiles of isoflavone composition since all the other compounds are equal. While in OSI the isoflavones are derived from soybean germ, in ORC they are derived from red clover. The major isoflavones from soybean germ are daidzein, glycitein, genistein, and genistin [25], while in red clover there are formononetin, biochanin A, daidzein, and genistein [26].

Isoflavones are phytoestrogens whose structure resembles that of the endogenous 17β-estradiol. Due to this structural similarity, these compounds can interact differently with the estrogen receptors (ERs) ER1 (predominant in the female reproductive system) or ER2 (predominant in males’ prostate and bones). This estrogenic activity induces hampered fertility depending on the dose and the organism’s maturation, an already known effect in the literature [26,27,28]. The zebrafish is a very sensible model to detect fertility alterations, and other compounds that were reported to cause fertility issues attributed to endocrine alterations were bisphenol S and 4t-Octylphenol [29,30]. However, it is essential to note that Ormona SI^®^ and Ormona RC^®^ are intended for menopausal and postmenopausal women, a condition where isoflavones positively affect complications [3,4,26,31].

## 4. Materials and Methods

### 4.1. Material Test

The nutraceutical products, Ormona^®^ SI and Ormona^®^ RC, were kindly provided by the company Ages Bioactive Compounds (Lot. 001, 2021 and 003, 2021). They were assessed for their δ-tocotrienol and geranylgeraniol content using gas chromatography coupled with mass spectrometry (GC-MS) and isoflavones (HPLC). A detailed composition profile is available in the patent process BR 2022 008408-4. Ten per cent of the formulations were composed of *Euterpe oleracea* (açaí) extract, previously characterized by our group [20].

### 4.2. Animals

This study used AB wild-type adult zebrafish (*Danio rerio*) purchased from the company Piscicultura Power Fish (Itaguaí—RJ, Brazil), and both sexes were used. The animals were kept in the Zebrafish Platform of the Drugs Research Laboratory (UNIFAP—AP, Brazil). The animals were habituated to the lab over 40 days with a 12/12 light/dark cycle (from 7 a.m. to 7 p.m.), and fed with ration (Alcon Colors—SC, Brazil) twice a day. The water system was maintained with controlled pH (6.0–8.0), conductivity (8.2 ± 0.2), and water circulation with treatment. All the procedures used in this study were approved by the Ethics Committee of Animal Use from the Federal University of Amapá (Brazil), under number 002/2022.

### 4.3. Oral Acute Toxicity

The acute oral toxicity was tested through the limit test following the OCDE 425 recommendations [32], with minor adaptations. The animals were separated into groups (*n* = 5), fasted for 24 h, weighed, and treated according to their groups. The oral treatment was performed with a micropipette, as described in [18]. OSI and ORC were administered at 500, 1000, and 2000 mg/kg, while the control group was treated only with the vehicle (2% DMSO).

After the gavage was performed, the animals were observed for behavioral changes, including swimming activity changes, tail tremors (Stage 1); circular swimming and posture loss (Stage 2); clonus, motility loss, and death (Stage 3) [23]. The animals were considered dead with a lack of response after a mechanical stimulus. After the end of observations, the animals were anesthetized through anesthetic culling following the American Guidelines from the Veterinary Medicine Association for Animal Euthanasia [33].

### 4.4. Histopathology

Histopathological analysis was performed on the animals’ livers, intestines, and kidneys. The animals and groups were the same as those used in the oral acute toxicity test. After being sacrificed, the animals were fixed in Bouin solution for 24 hours, decalcified in 7% EDTA for 48 h (Sigma Co., São Paulo, Brazil), dehydrated in crescent concentrations of ethanol (70, 80, 90, 100%), diaphonized with xylol, and embedded in paraffin. The samples were sectioned (6 µm) with a rotary microtome (CUT 6062, Slee Medical, Germany) and finally stained with hematoxylin and eosin, as described in [23]. The slides were observed with an Olympus BX-41 microscope, and pictures were taken with an MDCE-5C digital camera.

From the slides’ pictures, the histopathological changes index (HCI) was calculated. Each tissue change can be classified into I, II, or III according to the degree of severity. From the observed tissue changes, the following equation was used to calculate the HCI:(1)$$I=\frac{\sum_{i-1}^{na}ai+10\;\sum_{i-1}^{nb}bi+10^{2}\sum_{i-1}^{nc}ci}{N}$$
where *a*, *b*, *c* represent first, second, and third stage changes, respectively; *na*, *nb*, *nc* are the number of first, second, and third stage changes; finally, *N* is the number of fishes assessed per treatment [17,34].

From the HCI, it is generally accepted that values below 11 represent a healthy tissue with no significant changes, while values from 11 to 20 are considered low to moderate tissue changes, values from 21 to 50 are considered moderate to severe tissue changes, values > 50 are considered very severe, and values > 100 are irreversible.

### 4.5. Reproductive Toxicity

After being acclimatized, the animals were randomly divided into four groups: a naive control group of animals without treatment, a vehicle control group treated only with 2 µL of 2% DMSO, and two groups treated with OSI or ORC at 200 mg/kg. Each group had two male fish for each female fish, with *n* = 6 per group.

One week before beginning the treatments, the animals’ baseline fertility was evaluated by counting the deposited eggs and observing them for survival and teratogenesis up to 96 hpf. For this, after mating and spawning, the eggs were counted, washed, and put in Petri dishes with daily changed system water (70%) and kept in an oven (28 ± 2 °C) up to 96 hpf, as described in [35].

Then, oral treatment was performed over 21 days, according to the group. The number of deposited eggs, mortality rate, and teratogenesis were re-evaluated during this period. The hatched embryos were classified according to the severity of malformations observed (cardiac edema, tail malformation, scoliosis, yolk edema, growth retardation) or lethality. The observations were performed at 24, 48, 72, and 96 hpf. For each group, 10 embryos were photographed laterally and measured using Scion Image software.

### 4.6. Parental Gonad Histopathology

After the treatment period (21 days), the animals were euthanized and prepared for histopathology as described previously. For each group, the gonads of two female and four male animals were used, with 10 slides analyzed per animal [36].

In the females, the ovary follicle stage was classified into oogonium (Oo), perinucleolar oocyte (PO), cortical alveolar oocyte (CAO), early vitellogenic oocyte (EVO), or late mature oocyte (LMO). In males, the testes were classified into spermatogonium (Sg), spermatocyte (Sc), spermatid (Dt), or spermatozoon (Sz) [36].

### 4.7. Statistical Analysis

The results were given as the mean ± SEM. To evaluate statistical differences among groups, the Kruskal–Wallis test was used followed by the post hoc Dunn test in case of statistical significance. The significance level considered was 5% (*p* < 0.05). All analyses were performed with the software GraphPad Prism^®^ (5.03).

## 5. Conclusions

In the acute toxicity assessment, Ormona^®^ SI did not elicit any lethal or sublethal effect in the model used, and it was not possible to calculate the LD_50_ up to 2000 mg/kg. Moreover, only mild reversible tissue changes were observed, even in animals treated acutely with the highest dose. However, impaired fertility and embryo death in the animals were observed during the continuous treatment with Ormona^®^ SI and Ormona^®^ RC at 200 mg/kg, which the phytoestrogens from the formulation can explain. However, considering that the product is intended for menopausal and postmenopausal women, these phytoestrogens are beneficial. This research brings preliminary safety information about two new nutraceuticals, which should be further assessed in mammals.

## Figures and Tables

**Figure 1 pharmaceuticals-15-01434-f001:**
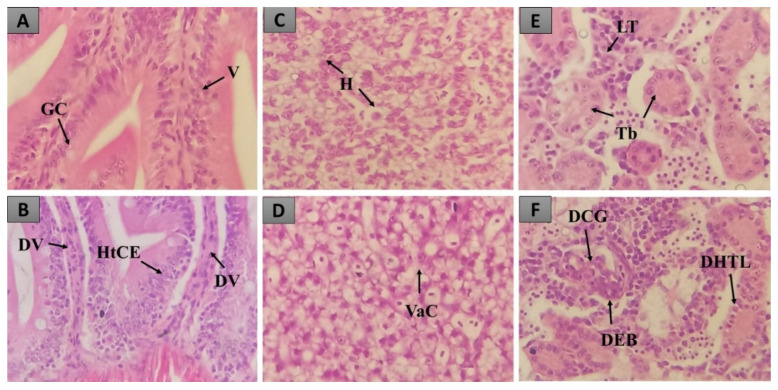
Representative pictures of the organs analyzed. (**A**,**B**) The intestine; (**C**,**D**) the liver; (**E**,**F**) the kidney. GC: goblet cells, V: villi, DV: dilation of vessels, HpCE: hypertrophy of the epithelial cells, H: hepatocytes, VaC: cytoplasm vacuolization, LT: linfoid tissue, Tb: tubules, DCG: glomeruli capillary dilation, DEB: decreased Bowman space. H&E (40×).

**Figure 2 pharmaceuticals-15-01434-f002:**
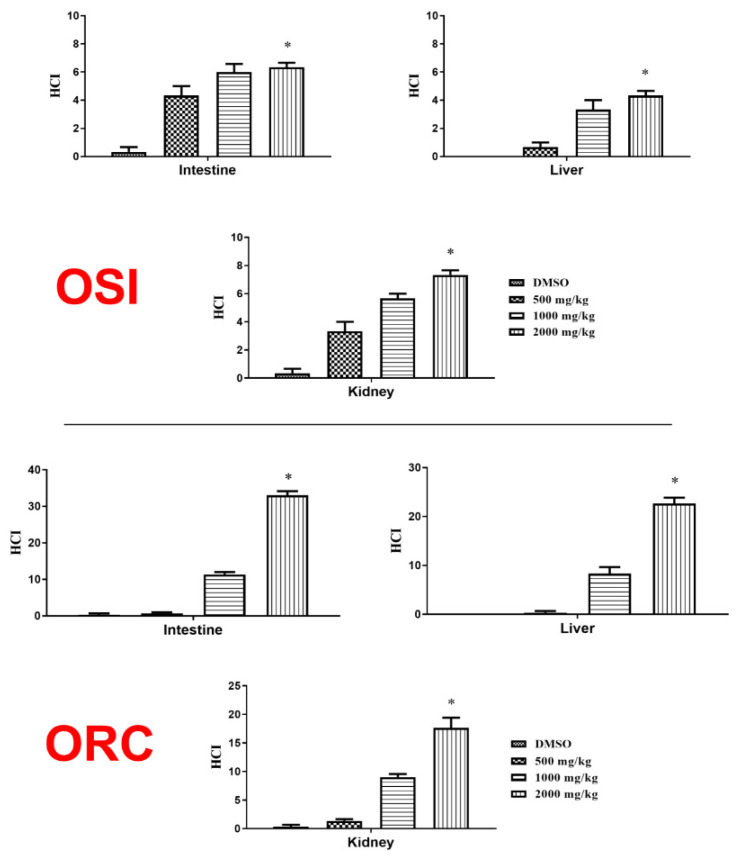
Effect of oral treatment with Ormona^®^ SI (OSI) and Ormona^®^ RC (ORC) (500 mg/kg, 1000 mg/kg and 2000 mg/kg) on histopathological change index in the intestine, liver, and kidney of adult zebrafish. The results are represented as a mean ± SEM. *: *p* < 0.05 vs. DMSO (Kruskal–Wallis followed by Dunn’s post hoc test).

**Figure 3 pharmaceuticals-15-01434-f003:**
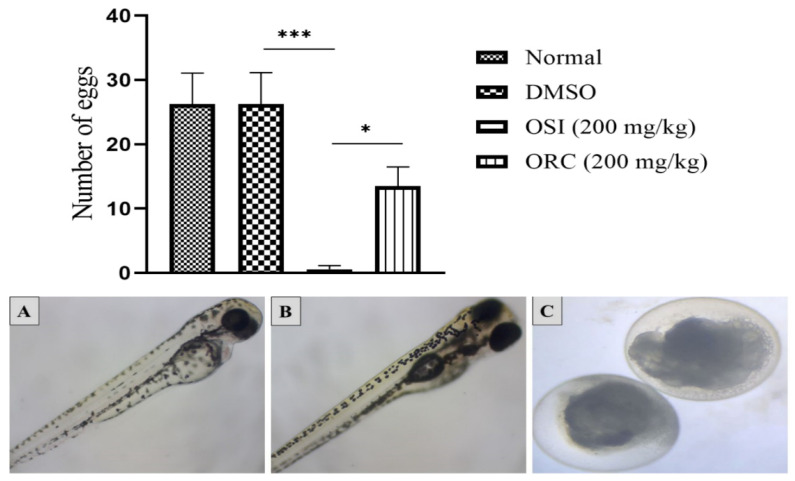
Top: mean ± SEM of collected eggs during 21 days of treatment (six parent fish per group); ***: *p* < 0.001, *: *p* < 0.05 (one-way ANOVA followed by the post hoc Tukey test). Bottom: (**A**,**B**) Representative picture of embryos from the untreated group and control groups (2% DMSO); (**C**) coagulated eggs from parental fish treated with Ormona^®^ SI (OSI) or Ormona^®^ RC (ORC) (200 mg/kg).

**Figure 4 pharmaceuticals-15-01434-f004:**
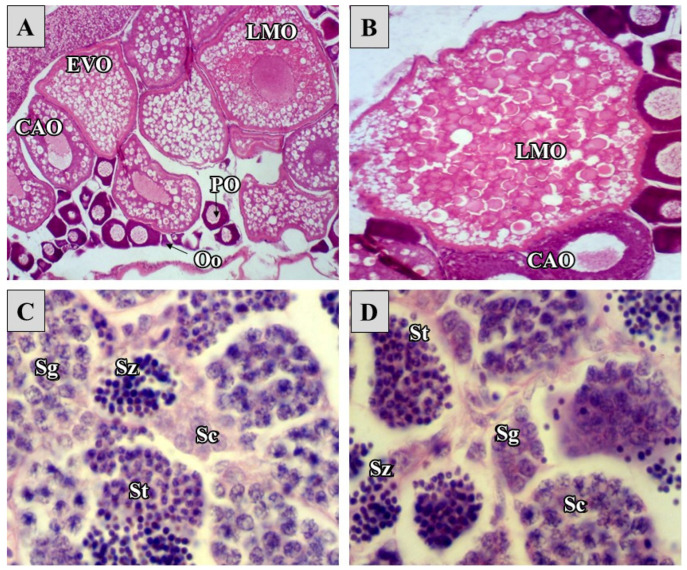
Representative pictures from the animals’ gonads. **Top**: ovary; **bottom**: testes. Oogonium (Oo), perinucleolar oocyte (PO), cortical alveolar oocyte (CAO), early vitellogenic oocyte (EVO), late mature oocytes (LMO). Spermatogonium (Sg), spermatocyte (Sc), spermatid (Dt), spermatozoon (Sz). (**A**,**C**) Representative picture from the untreated group and control groups (2% DMSO); (**B**,**D**) treated with Ormona^®^ SI (OSI) or Ormona^®^ RC (ORC) (200 mg/kg). H&E, 40×.

**Figure 5 pharmaceuticals-15-01434-f005:**
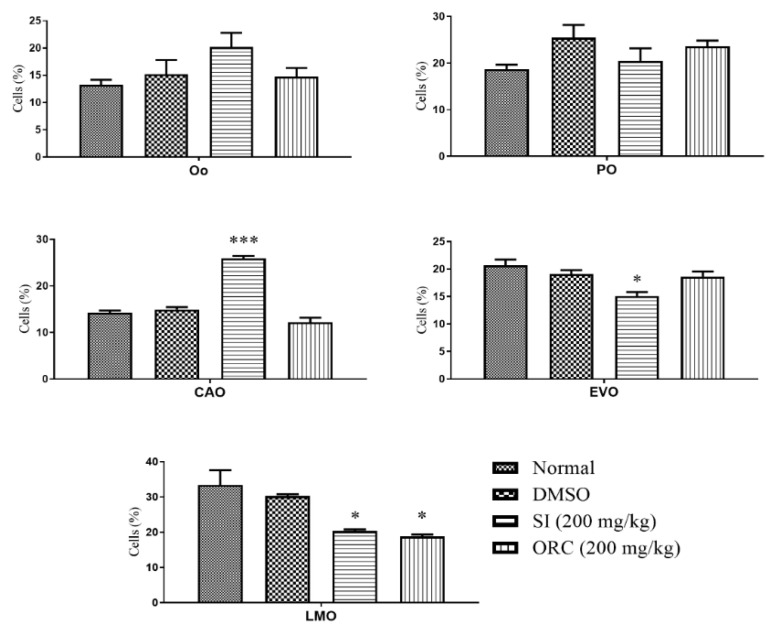
Effect of oral treatment with Ormona^®^ SI (OSI) and Ormona^®^ RC (ORC) (200 mg/kg) on frequency of females’ follicle stage (mean ± SD). ***: *p* < 0.001, *: *p* < 0.05, one-way ANOVA followed by the post hoc Tukey test. Oogonium (Oo), perinucleolar oocyte (PO), cortical alveolar oocyte (CAO), early vitellogenic oocyte (EVO), late mature oocytes (LMO).

**Figure 6 pharmaceuticals-15-01434-f006:**
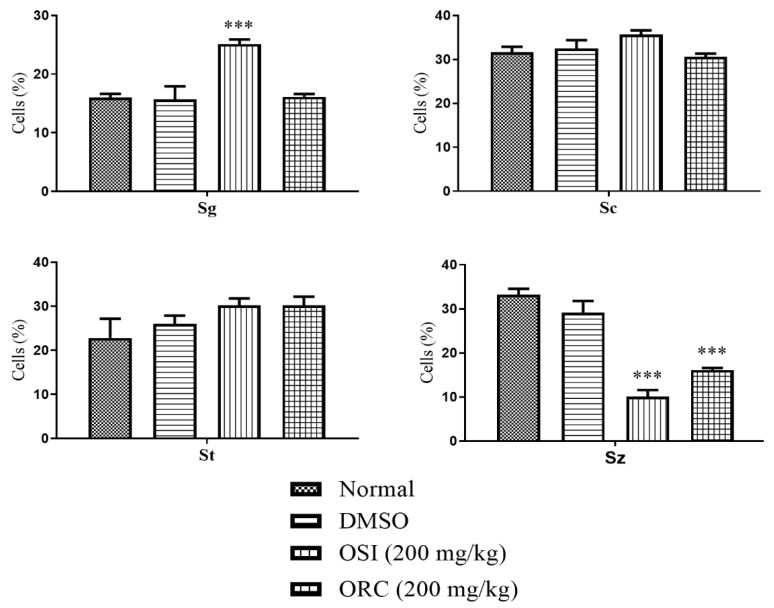
Effect of oral treatment with Ormona^®^ SI (OSI) and Ormona^®^ RC (ORC) (200 mg/kg) on frequency of males’ stages of reproductive cells (mean ± SD). ***: *p* < 0.001, one-way ANOVA followed by the post hoc Tukey test. Spermatogonium (Sg), spermatocyte (Sc), spermatid (Dt), spermatozoon (Sz).

**Figure 7 pharmaceuticals-15-01434-f007:**
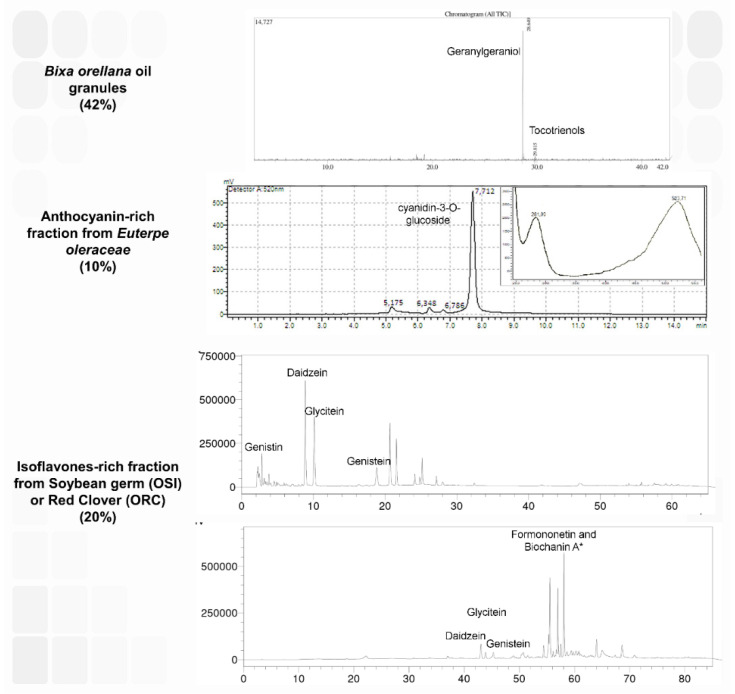
Profile of High Performance Liquid Chromatography (HPLC) analysis of the components of the nutraceuticals Ormona^®^ SI (OSI) and Ormona^®^ RC (ORC). The data and methods of evaluation have been published and can be found in [7] (*Bixa orellana* granules), [20] (*Euterpe oleraceae* extract), and [1] (isoflavone-rich fractions). *: It is presumed that two of these peaks represent formononetin and biochanin A based on the literature about red clover.

**Table 1 pharmaceuticals-15-01434-t001:** Effect of the treatments on the behavior and deaths.

Group (*n* = 5)		Stage I	Stage II	Stage III	Death
SI	2000 mg/kg	1/3	0/2	3/4	2
	1000 mg/kg	1/3	0/2	2/4	2
	500 mg/kg	1/3	0/2	0/4	0
RC	2000 mg/kg	1/3	0/2	2/4	2
	1000 mg/kg	1/3	0/2	2/4	2
	500 mg/kg	1/3	0/2	0/4	0
Control (DMSO)		1/3	0/2	0/4	0

## Data Availability

Data is contained within the article.

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
