# Peer review of "Acute and Reproductive Toxicity Evaluation of Ormona® SI and Ormona® RC—Two New Nutraceuticals with Geranylgeraniol, Tocotrienols, Anthocyanins, and Isoflavones—In Adult Zebrafish"

_pharmaceuticals, 2022, doi:10.3390/ph15111434_

Round 1

Reviewer 1 Report

This is a very interesting research article. The author conducted Ormona SI ® (OSI) and RC ® (ORC) with zebrafish. The evaluation of acute toxicity and reproductive and developmental toxicity of two herbal products shows the author's good ability to use zebrafish for health related products, especially the slides reading, analysis and display of histopathology, which leaves a deep impression on people. However, I have many suggestions and different views on the interpretation and analysis of the results. The specific contents are as follows:

1.       Is it appropriate to directly display the trade name for research articles? If it is academic discussion type, especially there are still inconsistent products in the analysis of conclusions, I personally think it is inappropriate, especially for magazines, so I suggest that the author modify the name of the product appropriately.

2.       P2,line74-75,the authors mentioned: Due to the insufficient mortality observed, even with high doses, it was not possible to calculate the LD50. However, the absence of LD50 does not mean that there is no toxicity, and according to the ICH guidelines, acute toxicity is mainly characterized by toxic reactions at specific doses. In this paper, higher doses of OSI and ORC have observed toxic reactions in acute toxicity and histopathology. It cannot be explained that OSI and ORC are safe because no toxic reactions occur at 500 mg/kg. Because the biggest problem with zebrafish testing is that there is no standard coefficient for conversion of zebrafish and human doses, unless there is data on in vivo exposure for conversion.

3.       On the contrary, for the evaluation of reproductive toxicity, there are few indicators and quantities, and there is only a single dose. As the author mentioned: The spacing in animals treated with OSI at 200 mg/kg increased rapidly compared to the control groups ORC at 200 mg/kg could decrease the hatching rate as well, alter the difference was not statistically significant compared to the control. Therefore, the hatchability is a key indicator of positive results.

4.       We know that  it is not the better the higher of 17 β-β-estradiol. On the contrary, oral estrogen may cause a series of side effects, such as breast pain, weight gain, hypertension, gallstones and liver dysfunction. At the same time, the contraindications of estrogen use also include not using estradiol during pregnancy. Systemic use of estrogen may lead to fetal malformation, and vaginal use of estrogen should also be noted. Therefore, the content and analysis of related substances are crucial. Moreover, it is also necessary to evaluate and analyze the safety of the product in combination with rats or other standard animals.

So I think this is a very good research article, and the results expose and prompt Ormona SI ® (OSI) and RC ® (ORC) risk needs to be further confirmed in combination with other standard drug toxicological animals (such as rats and rabbits), rather than just stopping now.

Author Response

The answers can be found in the attached letter.

Reviewer 2 Report

The study presents a good evidence for the use of isoflavones as a supplement in post menopausal symptoms. The manuscript is well structured and data are clear and well presented. However, a through language editing and rephrasing in some parts is required for more clarity. For figure 4, please change the black color for the symbols on the photos to another brighter color  to be easy to see.

Author Response

(The authors gave the same response as above.)

Round 2

Reviewer 1 Report

I think the author has made good modifications and basically reached the acceptance level. As for whether to change the trade name of the title, it is up to the magazine editor.